# Uncovering community needs regarding violence against women and girls in southern Ethiopia: An explorative study

**Metasebia Admassu**[1]*, **Lenka Benova**[1], **Christiane Nöstlinger**[1], **Aline Semaan**[1], **Aliki Christou**[1], **Claudia Nieto-Sanchez**[1], **Marie Laga**[1], **Misganu Endriyas**[2], **Thérèse Delvaux**[1]

1 Department of Public Health, Institute of Tropical Medicine (ITM), Antwerp, Belgium, 2 Southern Nations Nationality People's Region, Regional Health Bureau, Hawassa, Ethiopia

* metasebiaj@yahoo.com, ruhama.admassu@gmail.com

**Data Availability Statement:** The data cannot be disclosed because the processing of personal data

## Abstract

### Background

Violence against women and girls (VAWG) is a significant global public health problem and a violation of human rights experienced by one in three women worldwide. This study explores community perceptions of and responses to VAWG and challenges in accessing support services among female violence survivors in Arbaminch City.

### Methods

We adopted a phenomenological explorative qualitative study design. A total of 62 participants including female violence survivors, religious leaders, service providers, police, women, and men in participated in interviews, focus group discussions, and observations in August 2022. Participants were selected purposively, and the findings were analyzed thematically. We applied data source and respondent triangulation to increase the findings' trustworthiness.

### Results

Community perceptions of VAWG, specifically of intimate partner violence (IPV) and non-partner sexual violence (NPSV), varied depending on gender, age, and social position. IPV and NPSV were normalized through tolerance and denial by young and married men, while resistance to all forms of violence was common among women. Survivors of violence responded to the act of violence by leaving their homes, separating from their husbands, or taking harsh actions against their husbands, such as murder. Support for VAWG survivors was available through health care, free legal services, and a temporary shelter. Yet factors ranging from individual to societal levels, such as fear, lack of knowledge, lack of family and community support, and social and legal injustice, were barriers to accessing existing services. Nonetheless, violence survivors desired to speak about their experiences and seek psychosocial support.

needs to comply with stringent EU Data Protection Regulations, such as the GDPR. However, a reasonable request for accessing the data can be sent to the ITM Data Access Committee via the email: ITMresearchdataaccess@itg.be. Data can be made available for secondary research after review, approval, further anonymization, and signing of a Data Sharing Agreement.

**Funding:** The research was funded by Foundation de Luxembourg. Beyond the funding support the funder had no direct involvement in the study design, data collection, analysis and interpretation of the data or writing the manuscript.

**Competing interests:** The authors have declared that no competing interests exist.

## Conclusions

Our qualitative evidence gathered here can inform tailored VAWG prevention and response services such as interventions to shift social norms and the perception towards VAWG among different population group through raising awareness in schools, health care settings, faith-based venues, and using social media.

## Introduction

Violence against women and girls (VAWG) is a significant global public health problem and a violation of human rights [1, 2]. VAWG is experienced by one in three women worldwide, it has many forms, our study focuses on Intimate Partner Violence (IPV) and Non-partner Sexual Violence (NPSV) which are widespread and common forms of VAWG [3]. 37% of IPV cases were reported in the World Health Organization (WHO) African, Eastern Mediterranean, and South-East Aria regions. Where Africa accounts for 36.6% of the lifetime prevalence of physical and/or sexual IPV among ever-partnered women and 23.2% reported from high-income countries [1]. Although VAWG is common and widely spread, several studies have highlighted that it is under-reported with sexual violence being the most underreported form of violence [2, 4]. This is especially the case in lower and middle-income countries, where disclosing the experience of violence poses a great social and health risk to the survivor, thereby making it challenging to identify the true prevalence of all forms of violence [2]. In line with this, the prevalence of reported VAWG also varies within the same country across different regions. For instance, according to the Ethiopian Demographic Health Survey (EDHS), the prevalence of spousal violence varied between 9% in Afar to 38% in the Oromia region in 2016 [5]. The 2021 annual report of Southern Nations, Nationalities, and Peoples' Region Regional Health Bureau(SNNPR-RHB), Ethiopia, showed that violence cases were higher in some parts of the region and lower in others (SNNPR-RHB annual report [unpublished]). For instance, according to the regional report, the number of VAWG cases at the Arbaminch one-stop center accounted for 30% of the total cases reported in the region (SNNPR-RHB annual report [unpublished]).

The Ethiopian government has committed to addressing issues related to VAWG by creating a conducive legal and policy environment to promote the rights of women and girls [6, 7]. Ethiopia's government has put several laws and policies at all levels of the legal system in place that address fundamental human rights, including the adoption of the revised family law (2000) and the revised criminal code (2005) [6]. Ethiopia is also committed to eliminating all forms of violence from the private and public spheres to achieve Sustainable Development Goal 5(SDG-5) which addresses gender equality and women empowerment [8]. Despite such commitments, there are gaps: for instance, the criminal code of Ethiopia does not recognize marital rape as a criminal offense, and as a result, marital rape is not presented to the court [9]. Moreover, such form of VAWG is usually settled culturally through mediation which is an informal legal structure led by village (lower administrative structure in Ethiopia) level elders to resolve conflicts [10]. In SNNPR, as part of formal prevention and response services towards VAWG, nearly 14 one-stop centers and one safe house were established to provide services to violence survivors and to prevent VAWG(SNNPR-RHB annual report, 2021 [unpublished]).

There is substantial evidence available both in published and grey literature about the prevalence, root causes, consequences of VAWG, help-seeking behaviors, and survivors' health-seeking behaviors. However, there is a shortage of context-specific evidence on how

communities perceive VAWG and the preventive measures and response services that exist. Additionally, there is a need to explore how to improve VAWG prevention and response services from the perspectives of violence survivors and community. The WHO report stated numerous complex factors contributing to the occurrence of VAWG, through the complex interaction of factors ranging from individual to societal level [3]. In line with this, risks associated with violence are not evenly distributed, in terms of e.g. geographic region, age, and gender, which makes the issue of violence complex and stresses the need to understand the context in which it occurs. Studies also show that assessing communities' (the people residing in a specific geographic area and sharing the same values and cultures) [11] health needs are key to inform possible recommendations to local planners and implementers in the study region for future improvement [12]. Hence, we aimed to explore perceptions of, challenges and gaps in the existing response and prevention services, and identify violence survivors' and other community members' additional needs to inform future prevention and response towards VAWG in Ethiopia.

## Methods

### Study design

This study adopted a phenomenological explorative qualitative research design using in-depth interviews (IDI), informal conversations (IC), focus group discussions (FGD), and observations as data collection techniques.

### Study setting

We conducted the study in Arbaminch City, Gamo zone, SNNPR Ethiopia (Fig 1). SNNPR is one of 11 administrative regions in Ethiopia and Gamo zone is one of the 11 zones and 7 special districts in SNNPR. Arbaminch City is the administrative capital of Gamo zone located 505 km south of Ethiopia's capital, Addis Ababa. An estimated 125,562 people live in Arbaminch city, which has one general hospital (Arbaminch General Hospital), five primary hospitals, and 57 health centers (SNNPR-RHB working document, 2023 [unpublished]). The hospital hosts the one-stop center and works closely with a stand-alone safe house.

### Study population

The population of interest were female violence survivors, men, women of reproductive age (15–49 years), religious leaders, elders, and VAWG service providers including health, legal, and other related services (Fig 2).

### Participant and site selection

**Site selection.** We utilized purposive sampling with the help of healthcare providers and a review of regional documents to identify study sites and study participants. We selected the Gamo zone as our study site since it had a higher number of reported violence cases than the rest of the zones in the region during the data collection period, with most of them from Arbaminch city (SNNPR-RHB annual report, 2021[unpublished]). In addition, the city has a one-stop center and a safe house for violence survivors, making it an ideal location for our study. We chose Kebele six within the zone because it had the highest number of reported violence cases in the one-stop center client registry book.

**Participant selection.** The study's participants for the FGD consisted of men and women who were either married or cohabiting, and had been with their current spouse or partner for at least a year. Women of reproductive age and men who had resided in Arbaminch City for at

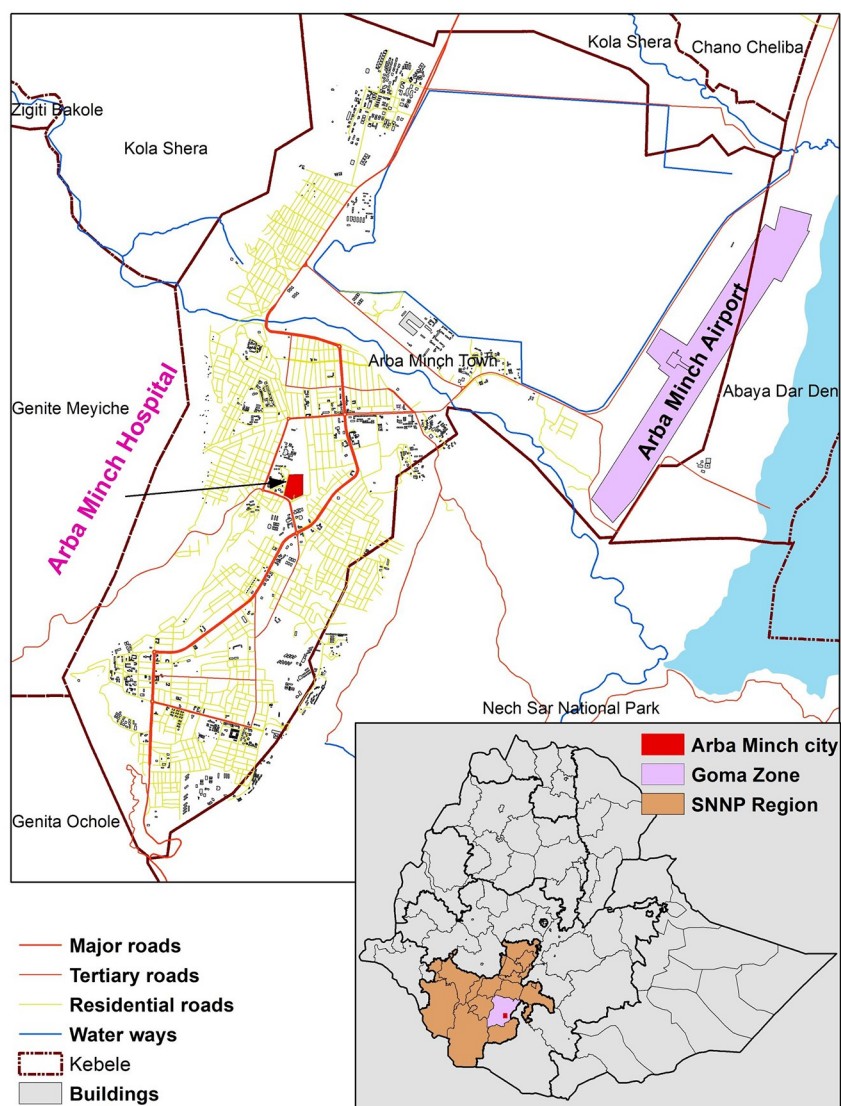

**Fig 1. Map of the study site in Southern Ethiopia, Gamo zone, Arbaminch City, 2022.** The figure was created in ArcMap V.10.5 (ESRI, Redlands, California, USA) by the author with data sourced from OpenStreetMap.

least six months, along with service providers on VAWG, were interviewed individually. Violence survivors were selected based on the criterion that they were under care at the safe house and the one-stop center during the data collection period. Data collection was carried out until we reached saturation, which means that participants shared no new information. However, the number of survivors of violence included in the study was limited to those who were in care at the safe house and the one-stop center at the time of the study. Two female survivors declined to participate.

## Data collection tools

**IDI.**   A one-time IDI was conducted with each study participant except for female violence survivors. We used 4 different interview guides depending on the respondent type (S1 Annex).

Given the sensitive nature of the topic, IDI interview guides for female violence survivors were structured in a way that they could be administered over a course of 2–3 days visits to be

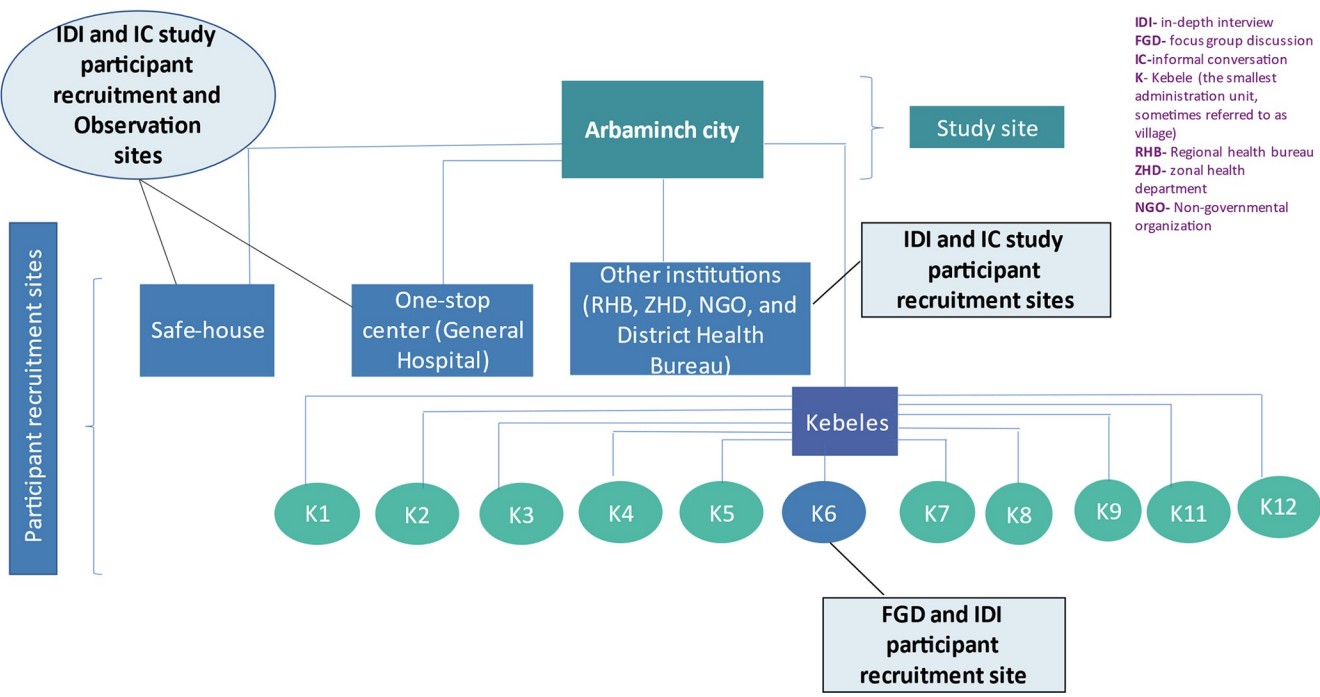

**Fig 2. Illustration of study and participant recruitment site.**

able to establish good rapport and trustful relationships. For the remaining IDIs the average interview duration was 45mins.

**FGD.** All FGDs were conducted using a semi-structured FGD guide. FGD questions addressed the study objectives and explored topics related to women's safety and security in the community, community perceptions of VAWG, to identify available response and prevention services in the community, barriers to accessing VAWG services and additional services to reduce VAWG. The average time for the FGD session was 1hr:10mins.

**IC.** Was conducted at the safe-house and the one stop center, with violence survivor, service providers and violence survivor relatives which were not included either in the IDI or FGD. The research team asked questions relevant to the study objective.

**Observations.** Site observations were conducted to understand the overall response services. Initially we prepared a structured guide for the observation. However, the research team agreed not to use the guide, so that observations would not be limited and so we could obtain as much information as possible. The observation at both sites took on average 6 days.

The data collection guides were adopted from validated instruments used in previous studies on VAWG [13–15]. All data collection tools were translated to the local language, Amharic, by the native Amharic-speaking PI.

## Data collection

Data collection took place in August 2022. Two public health professionals collected the data. The PI provided training to the research assistant to ensure data quality, informed consent, issues of privacy and confidentiality, and data protection and security. Data were collected in a convenient private location and face-to-face except for one IC which we conducted by telephone.

Interviews with female violence survivors were formulated to minimize inducing stress, and prior communication and confirmation about each violence survivor was confirmed from the case managers to ensure survivors felt comfortable to participate in the study. Moreover, the data collection team confirmed availability of backup psychological and health service support to survivors in case they raise any untreated illness or trauma during the interviews.

## Data analysis

The audio files were simultaneously transcribed and translated in to English and language consistency and clarity were checked and corrected by the PI and the research assistant. As a first analytical step, for the purpose of uncovering the meaning of the data, field notes and memos written during and after interview and FGD were coded by the PI and the research assistant (ME) and similarities and differences were discussed during the time of data collection. The transcribed data including notes from our observation and IC were imported into Dedoose software and were coded for further analysis by the PI. In the second analytical step, three research team members (AS, AK, and TD) coded the transcripts inductively, based on inductive reflexive thematic analysis approach [16] by identifying and organizing data pertaining to each of the study objectives. Similar codes were combined and the team developed broader themes and sub-themes (S1 File). As part of the final step (deductive step), some key findings were mapped onto the structure of the Socio-Ecological Model (SEM), a theoretical model widely used in violence prevention [17].

## Ethics

Ethical approval for this study was obtained from the Institute of Tropical Medicine Institutional Review Board and SNNP-RHB. We sought written consent was sought from all IDI and FGD participants and oral consent for IC and observation before commencing the study.

## Reflexivity, positionality, and validity

The PI is a female researcher at the Institute of Tropical Medicine who attended the Qualitative and Mixed Methods in International Health Research course. A male research assistant(ME) was a Ph.D. fellow at Hawassa University at the time of data collection. They both are Ethiopians and fluent in Amharic. To facilitate smooth and open conversation, the discussions and interviews with women were conducted by the female researcher, while the interviews with most men were led by the male research assistant.

As an Ethiopian woman, interviewing and reading the transcripts of violence survivors and women was a traumatizing experience. My position as an Ethiopian woman was impacting my analysis: given the emotional affect, I hesitated at times to clarify some of the findings to not burden the participants again with data collection. The entire data collection process took longer than anticipated due to the coping process with the at times shocking content participants shared during data collection. Nevertheless, continuous support, discussion, and feedbacks from other co-authors who are non-Ethiopian helped me to clarify my position and helped me to bring this perspective into the analyses in a methodologically sound manner.

The study was reported in accordance to the consolidated criteria for reporting qualitative research (COREQ) Checklist [18].

## Inclusivity in global research

Additional information regarding the ethical, cultural, and scientific considerations specific to inclusivity in global research is included in the S2 File.

**Table 1. Types of study participants and method of data collection.**

| Type of study participant | Actual study participant | IDI | FGD# of participants (# sessions) | IC | Total study participant (# sessions) |
|---|---|---|---|---|---|
| **Women and girls** | Violence survivors | 3 | | 1 | 4 |
| | Married women and girls | | 8 (1) | | 8 (1) |
| | Unmarried women and girls | | 10 (1) | | 10 (1) |
| | Girls with no violence experience | 3 | | | 3 |
| **Men** | Unmarried men | | 9 (1) | | 9 (1) |
| | Married men | | 10 (1) | 1 | 10 (1) |
| **Health professionals** | Nurses | 1 | | | 1 |
| | Doctor | 1 | | | 1 |
| | Psychologist | 2 | | | 2 |
| | Health officer | 1 | | | |
| | Health extension worker | 1 | | | |
| **Other service providers** | Police officer | 1 | | | |
| | Lawyer | | | 1 | |
| **Other community members** | Religious leaders | 5 | | | |
| | Elder | 1 | | | |
| **Officials** | NGO officer | | | 1 | |
| | Women and gender officer at the zonal level | | | 1 | |
| | Women and gender officer at the district level | 1 | | | |
| **Total** | | **20** | **37 (4)** | **5** | **62 (4)** |

## Results

We included 62 participants (Table 1). During the FGD with married women, one participant dropped out for personal reasons not associated with the study.

We identified five key themes spanning perceptions of and response to violence; factors contributing to violence and recommendations for VAWG prevention and response.

### Theme 1: Community perceptions and understanding of VAWG

Perceptions of VAWG varied by different participant types. Not all forms of violence were taken seriously; participants reported conditions when VAWG was tolerated or deemed unacceptable (Table 2). Almost all women and violence survivor participants mentioned that any form of violence is unacceptable: "*...Let alone being beaten, I don't want to be insulted, it is difficult for me to bear. It will not give you confidence if your husband insults you and gives you a low place in the house.*" **MW_FGD.**

VAWG within marriage was commonly considered as minor, common, and disagreement. Most participants did not identify IPV as a common form of violence in the community and referred to IPV as "*we hear a lot of fight among married people, but as long as they are married they will sort it out*". However, some participants acknowledged the need to accuse the perpetrator if VAWG in a marriage left the victim with visible wounds.

VAWG to unmarried girls was considered serious if the survivor is humble, shy, underaged, and if the violence act leads to virginity loss. Respondents considered VAWG acceptable if survivors are outgoing, sex workers, or inappropriately dressed girls. Moreover, VAWG forms such as insults, controlling behavior, groping, grabbing, and attempted rape were perceived as normal.

**Unacceptable.** Young female participants understood and perceived VAWG from a human-rights perspective regardless of the traits of survivors and perpetrators. Health service providers, religious leaders, and participants engaged in VAWG prevention activities perceived VAWG as an unacceptable killing of the generation.

**Table 2. Illustration of tolerated and unacceptable violence perceived by study participants with some illustrative quotes.**

| Conditions | VAWG is tolerated | Illustrative quotes | VAWG is unacceptable | Illustrative quotes |
|---|---|---|---|---|
| **Forms of violence** | Forced sex[rape] in a marriage, physical violence without major visible injuries/wounds, attempted rape, insult | *"...in our community no one should say my husband raped me, this is shameful thing to say."* MM_FGD<br>*"...Sometimes, married women run out and shout without any physical harm."* MM_IC/FGD<br>*"...If it is an insult nobody will take it seriously, it is taken as normal"* HEW_IDI | Rape resulting in losing virginity, physical violence with major visible injuries/wounds and repeated physical violence | *"...If we see an injured part like the face or eye, we should report it to legal bodies."* MM_FGD |
| **Individual characteristics** | Playful, inappropriately dressed girl | *".. the girl's dressing style should not tempt a man."* MM_FGD | Shy, humble, calm | *"...The people around the neighborhood believed me, because they know I am a calm girl."* VS_IDI |
| **Place of violence** | Risky area | *"...If you report an act of violence committed in risky area, no one gives you an ear."* UMW_FGD | Sudden occurrence of violence on the way to [market, school] | *"...if the violence is unexpected like while she is returning from market, the community should support her."* UMM_FGD |
| **Time of violence** | At night after 8 and 10 | *"...People will just say, what was she doing at this time? they will not come out to help."* MM_FGD | | |
| **Occupation of the violence survivor** | Sex worker | *"...if a sex worker shouts for help no one will come out for her."* UMW_FGD | Domestic workers | *"...people should be advised not to abuse domestic workers and when they face violence it should be notified."* MM_FGD |
| **Age of survivor** | | | Children less than 15 years | *"...There was also a time where I defended a girl who was 14 or 15 year. I said to him, this is a crime! She is a kid how can you plan to do this act [sex] with this girl?"* MM_FGD |

**Minor, rare phenomenon and no VAWG.** Some male participants and Muslim religious leaders perceived IPV as minor and rare. Most married men denied the existence of VAWG. Men assumed that the absence of clear attempts by violence survivors to resist violence acts as though survivors were consenting: *"...violence could have been stopped if there was no willingness(acceptance) by women. Violence continues to exist because there is willingness. If a woman resists it, every one of us would have been in jail."* **UM_FGD.**

**Married women should expect VAWG.** For some married men, VAWG was perceived to be perpetrated towards unmarried girls only. Married men believed that married women should expect to experience some level of violence, particularly if the husband was a drunkard, if she spent the entire money that he gave her before the end of the month, if she stayed out without him knowing, argued with him, if she was a housewife, not educated, or didn't care for the children.

*"...assume a man earns 1000 Ethiopian birr and gives 900 birr to his wife, having 100 birrs for himself. If the wife spends the 900 birr right away how do you expect peace in the house? Don't you expect violence in the house?"* **MM_FGD**.

## Theme 2: Community and violence survivors' responses to VAWG

Response to the act of violence by different population groups depended on perceptions towards VAWG.

**Non-interference, stigma, and gossip.** For the tolerated types of violence (Table 2) some participants, mostly women, reported that community members usually gossiped, stigmatized,

did not interfere even if they saw or heard the act of violence, and were unwilling to witness. As one unmarried man's comment illustrates,

> "…when people hear a girl who overacts or dresses inappropriately was raped, they will say 'Hashu'(an expression meaning she deserves it)." **UM_FGD**

**Marry the rapist.** Many participants stated that families of violence survivors usually tolerated or stopped accusing perpetrators legally to maintain family honor, and in fear that the survivor will not find a future husband, leading families to force the survivor to marry the rapist.

**Silence, non-disclosure, seek care or help.** Some families or survivors did not disclose due to fear of gossiping and stigmatization. Lack of financial capacity or a family member's financial dependence on a perpetrator usually lead to not taking the case to court in exchange for agreement receiving compensation from the perpetrators: "…*Sometimes some parents take risks and fight for their rights but most of the time they discontinue the legal process and end the case through the customary mediation to avoid stigmatization from the community.*" **HEW_IDI.**

**"This is the way they are".** Most IPV cases do not get disclosed or IPV survivors usually don't seek help. In some scenarios when women shout for help, the neighbors usually do not interfere:

> "…*in our village, there is a woman, it is very common for her to shout for help every day. When she shouts no one comes out for her. The community say 'This is the way they are.'*" **MW_FGD**

**Humble and good wives.** There was also a notion that, if a married woman takes a case to court she will be blamed, whereas those who tolerate were described as "good and humble women":

> "…*Honestly, I see most women rush to divorce… Of course, I am not saying all women are like that. Some good women say, even if he makes a mistake let me shoulder him. There are such humble wives*" **MM_IC**

**Light action towards perpetrators.** Participants mentioned tolerance of perpetrators or weak sanctions against them. Perpetrators were begged, advised not to repeat the act, or asked to give compensation, or VAWG case was overlooked. However, survivors were advised to leave their hometown or tolerate:

> "…*She brings police, brings elders, but nobody solved her problem. So she left the home, and she never came back.*" "…*Most of the time we hear about a 7-year-old, or a 10-year-old girl raped. But measures are not adequate for those who commit the crime.*" **MM_FGD**

**Fighting for survivors, believing, and supporting.** Some community members including young men fought for survivors, stated that they will not tolerate this, and extended support when violence occurred.

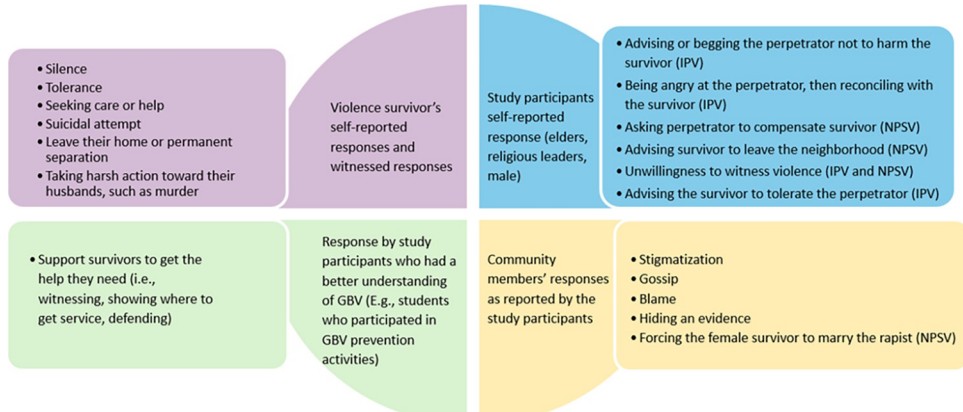

**Fig 3. Illustrations of responses to the acts of violence by the study participants.**

*". . .older male students deceive younger girls with a few things like tea invitations. Then they take them to the bush and rape them. We have once caught and reported such a case to the police."* **MM_FGD**

In the case of perceived unacceptable violence (Table 2), participants reported that the community initially believed survivors and were willing to report to the police or show sympathy to the survivor and extend their support. Fig 3 shows a range of responses by different community members to VAWG. Responses were determined by the forms of violence described in Table 2. In most cases, even if the acts of violence were perceived as serious by the community, the social and legal responses and actions towards the perpetrators were light or minimal.

### Theme 3. Existing services, access, and service delivery barriers towards VAWG

**Existing services.** Participants, especially service providers and those who engaged in providing technical support mentioned most of the existing services (S3 File). Prevention activities such as, school-based interventions were mentioned by two younger girls who participated in these activities. Community members reported response services such as police, women and children's affairs office, and customary mediation, situated in the kebele or village.

**Care-seeking experience.** Observations at the safe house show that all violence survivors who sought care were NPSV. Service providers at the one-stop center and at the safe house confirmed that the most common cases visiting the service center were survivors of physical and sexual violence among young girls. Service providers reported that care-seeking among IPV survivors was minimal. Most VAWG preventive activities were done in rural areas focused on schools. According to the service provider at the safe house, this was to create awareness among young girls and prevent internal migration to urban settings. Nevertheless, poverty pressured them to migrate to cities that hold higher risks for the occurrence of NPSV.

**Barriers to care seeking.** Several factors were mentioned as perceived and experienced barriers to accessing services, according to the SEM levels (Table 3). Due to these barriers, it was rarely reported that violence survivors seek help specially IPV survivors. Close friends and family members were the source of help for those who decided to seek it. Service providers reported that some violence survivors came to the healthcare facility after experiencing other health-related conditions such as becoming pregnant, or after acquiring HIV, stress, or when they felt they couldn't tolerate the violence and the negative consequences anymore. Interviews

**Table 3. Summary of findings using the SEM for perceived contributing factors for becoming a victim and perpetrators of violence, experienced barriers, and community suggestion.**

| SEM | Perceived contributing factors | Experienced barriers to help-seeking or disclosure | Suggested response activities by the study participants | Suggested Prevention activities by the study participants |
|---|---|---|---|---|
| Individual level | **Younger age** <br>**Female gender** <br>**Occupation of women (sex work, domestic worker)***<br>Substance abuse and alcohol drinking** | • Lack of information on VAWG services<br>• Not realizing the act of violence<br>• Fear of the perpetrator, blame, discrimination and stigma | • Speaking about or disclosing the experience of violence | Self-care (staying at home, avoiding traveling to risky areas)<br>Being submissive to a husband<br>Support to men with addiction |
| Relationship level | *Poor parent-child communication on VAWG*<br>*Strict parenting of daughters*<br>*Existing IPV among the parents*<br>*Separated/not-cohabiting parents**** | • Lack of support from parents or close relatives | • Enhance positive parent child communication<br>• Acceptance by family,<br>• Compensation (money, in-kind) to violence survivors by the perpetrator | Building good communication between parents and children, especially with a teenage girl<br>Awareness creation of how to raise children |
| Community level | **Lack of community support to violence survivors**<br>*Availability of marked risky areas (spots), where violent acts are common (e.g. riverside, bushes where women perform routine domestic tasks)*<br>*Lack of community awareness on VAWG* | • Lack of community support<br>• Weak customary mediation****<br>• Social stigma, discrimination | • Emotional support to violence survivors by the community<br>• Avoid customary mediation<br>• Condemn the act of violence by the community | Adopting cultural norms that empower women<br>Strengthening religious teachings on VAWG<br>Assigning armed men in the community to enhance safe movement for women at night<br>Taking action on risky places (hotel, bushes...) |
| Societal level | *Weak legal sanctions against VAWG perpetrators*<br>*Gender inequality*<br>*Lack of attention to violence related services by the government,*<br>*Modernization, and social media misuse*<br>*Poverty* | • Weak legal actions against VAWG perpetrators<br>• Need for presenting witness and evidence by legal body<br>• Lengthy legal process<br>• Strict social and gender norms<br>• Lack of money to access service | • Placing proper punitive action towards violence perpetrators<br>• Punishing the community that stigmatizes the survivor<br>• Enhance continuous emotional and psychological support to survivors<br>• Implementing interventions that promote disclosure Infrastructures (isolated laboratory and pharmacy) to maintain violence survivors privacy | Establishing youth, sports clubs, and libraries<br>School based programs like peer education, discussion and awareness creation on violence in and out of school<br>Economic support to women, and jobless men |

*Texts in bold show contributor factors for becoming a victim of violence

**Texts which are underlined show contributor factors for becoming a perpetrator of VAWG

***Texts in Italics show contributor factors for becoming a victim and a perpetrator

****Community mediation system for traditional justice [31]

with violence survivors confirmed that two of them were already pregnant by the time they sought formal care and one reported extreme distress due to the violence experienced and the barriers in seeking help.

**Barriers to service delivery.** Service providers and implementers mentioned lack of budget, shortage of resources, lack of coordination, integration, lack of advocacy, and weak referral linkages as challenges in delivering services.

## Theme 4. Perceived contributing factors to VAWG

Perceptions about contributing factors towards VAWG are presented according to the SEM levels (Table 3).

**Individual level.** Age, employment, education, low income, substance and alcohol use were reported by most participants as contributing factors to being either a violence victim or

perpetrator. Most participants reported that married women without a job, internal migrants, domestic workers, sex workers, and girls under 18 years were vulnerable, or pre-disposed groups to becoming violence victims. Women and girls reported that substance and alcohol abuse by men were key pre-disposing factors for becoming violence perpetrators, a factor which led women to perceive that violence was hopelessly unavoidable:

> "...men around this area use different addictive substances, as you are a female you can't escape from all men. One day you will be a victim." **MW_FGD**

Contrarily, male participants reflected that women's and girls' traits and actions contribute to them becoming violence victims. For NPSV, most men reflected that girls expose themselves to violence by overacting or dressing inappropriately as a result of modernization or social media use. Some married women also perceived inappropriate behavior by women such as alcohol use, and failure to self-prevent violence (avoiding travel to risky areas/at night, proper dressing style) as contributing factors for becoming a victim. For IPV, men mentioned that a wife's behavior (not being submissive) is a provoking factor for a man/husband to become abusive.

**Relationship level.** The most commonly mentioned perceived contributing factor was communication between parents and children and among partners in a couple. poor communication between parents and young girls was perceived to be a contributing factor for a girl to become a victim of violence. The discussion with married women emphasized that the restrictive behavior of parents especially mothers towards their daughters was one of the main reasons young girls leave their family house which might expose them to various risks including violence:

> "...my mom was very strict towards me. So, I decided to marry to escape from her nagging, but now I am also living in an abusive environment." **MW_FGD**

Different ways of raising children and their different life experiences were reported as contributing factors to becoming violence victims or perpetrators. For instance, girls are usually expected to stay in the house with their mothers helping out with domestic work while boys are encouraged to stay out and play with peers. Participants believed that such gendered differences prevented many girls from exercising their rights freely:

> "...when I grew up, if I stay out of the house for a minute it is a big deal, while my brothers can stay out of the house the whole day. This thing needs to be improved" **VS_IDI**

Poor communication between partners in a couple was portrayed to influence how their children will act in the future. Not having a stable and healthy relationship between couples was also raised as a contributing factor to a child becoming violence victim:

> "...I didn't grow up seeing love between my parents as I see in other houses...If there is no love between parents the children will not experience love and they will face violence" **VS_IDI**

**Community level.** The most mentioned factor was the presence of risky areas in the study setting including, bushes, river sides, dark neighborhoods, bars, and some specific kebeles which were mostly mentioned by women participants. The intensity and frequency of violence that occurs in these marked areas were well recognized, and participants pointed out that some women were raped by multiple gang members at once in these locations.

*". . .we hear many violence that happened in the bushes on girls who collect wood. Forget girls, even old women are raped there."* **UM_FGD**

**Societal level.**   Women mentioned that the availability of community-based customary mediation often led to little or no repercussions for the perpetrator's act of violence. Men in the community usually bring elders to mediate with their wives or partners after committing abuse, then according to the social norm a female violence survivor should also agree to this mediation. Women believed that such mediation was beneficial to perpetrators and contributed to increasing violence in the community.

*". . .I have a sister who lives with an abusive husband. Every time she runs away from home following the abuse, her husband sends elders. Then, she returns to him, because if she says no to him and remains with us, the community will blame her saying, why are you breaking your marriage?"* **UW_IDI**

Most participants reflected that failure to implement and interpret policies and laws correctly is a predisposing factor for VAWG. It was reported that, the legal system was corrupted and that the penalizing criteria favor perpetrators more than survivors. For example, needing a witness and evidence (physical, medical) to prove that the violence occurred, and the lengthy legal process were reported as barriers that discouraged violence survivors from seeking justice. Participants reported that, failure to seek justice and not getting the right decision as a contributor factor for VAWG.

*". . .even with the correct evidence majority of the time, you will not get justice. Violence cases will be covered up, you will go to a few steps then it stops, nobody gives attention to it"* **MW_FGD**

Violence survivors reported that it was traumatizing for them to be aware and see the perpetrators living their lives freely in the community, while they take the blame. Gender inequality was reported as an overarching contributing factor, as it gives high position for males while giving little attention to girls and women. This was reported to create a climate where some forms of violence are encouraged in the community.

## Theme 5. Community and violence survivors' recommendations towards prevention of and response to VAWG

In this theme, we summarized recommendations on the prevention and responses to VAWG (Table 3).

**Self-care.**   Violence survivors and some women participants suggested self-care to prevent VAWG. Some older and married women reflected that it is a girl's responsibility to prevent violence by avoiding travel to risky areas, dressing appropriately, and minimizing risky behavior such as avoiding alcohol consumption. Only a few women mentioned that a girl has a right to wear what she likes and could travel wherever she wants. Similarly married men also agreed on the self-care to the point where putting the responsibility of preventing VAWG on women:

*". . .Even if the husband is abusive, the wife knows her husband better than anyone else, therefore she should treat him properly, if she fails to do that, violence will not be reduced."* **MM_FGD**

**Good communication.** At the relationship level, most respondents including violence survivors, married women, elders, and men discussed the idea of investing in children-parent communication strategies and building good relationships to be the most useful strategy to protect girls from being abused and for preventing boys from becoming violence perpetrators:

"*. . .If girls are handled properly and they are strong, they can withstand the challenges they face and also don't hide the violence they face.*" **Elder_IDI**

**Disclosure.** All three survivors described the positive feelings following disclosure and receiving proper care. Other participants also highlighted the importance of disclosure, noting that it brings the case of violence to light, subsequently possibly influencing a decline in violence in the community:

"*. . .I felt relieved when I spoke about the violence. Nobody should suffer alone, if you have to die you must die after speaking about it. They may not get a fair decision at the legal sanction, but rather than keeping it secret, it is good to talk about it. It will give them peace of mind. . .. Holding inside is the biggest illness ever.*" **VS_IDI**

**Continuous care and support.** At the community level violence survivors, female participants, and elders mentioned the importance of having continuous care and support from institutional services and family, neighbors, and community members. Almost all survivors mentioned being accepted by family, community members was what they needed the most to recover from the psychological trauma and to reduce violence in the community.

"*. . .you know what, when a girl becomes a victim, it will be an issue for 2 and 3 weeks and she may get support during those times. After that, no one will talk about it. But the support should be continuous. She needs to have good treatment from the community, friends, and family.*" **UW_IDI**

**Social and legal justice.** At the societal level the importance of revising and putting proper punitive mechanisms on the perpetrator was shared by all participants as key to reducing violence and for survivors to recover from the psychological trauma they experienced. Many participants repeatedly reported that they perceived some activities which were mentioned as a response to violence rather as a primary preventive activity. For example, while the legal system's primary role is to respond to the violence survivor, almost all participants believed that proper translation and application of the law can contribute to preventing violence in the community.

## Discussion

This study explored VAWG in Southern Ethiopia, Gamo zone Arbaminch city. Our findings added the perspective of different population groups within specific communities about VAWG. We applied an intersectional SEM framework to explain how multiple factors impact violence survivors' safety and security, and make them vulnerable to VAWG, and influence help-seeking. The study also revealed the community and violence survivors' needs in averting VAWG in the community.

Our findings on the perceptions of IPV and NPSV showed a range of ideas that varied by age, gender, and the different roles participants had in the community. Female community members felt their day-to-day life was challenged by the dominant risk of VAWG in the community and the lack of safe spaces for them to feel protected from violence. The community primarily considered acts that cause visible physical injury to women as acts of VAWG. Some perceptions of IPV might be the result of overlooked actions by the Ethiopian criminal code which does not recognize marital rape [9]. This could explain why IPV cases were resolved culturally and not legally. This finding is in line with those of a systematic review conducted in Ethiopia and other studies [9, 19–22]. NPSV was also normalized based on certain criteria that the community set. This normalization was seen mostly from males, while women resisted any form of violence, and few young men and religious leaders were against VAWG. Despite this, women felt unsupported in most cases. Ultimately, out of frustration, most IPV and NPSV survivors remained silent or were tolerant towards violence, chose to leave their homes, separate from their spouses, or took harsh measures such as murdering their spouses in self-defense. This finding was in line with those from a study conducted in Ethiopia where women's reaction to IPV is being silent, tolerant, leaving home, self-defense, or seeking help [20, 23]. As with a study conducted in Malawi, nearly 34% of all respondents held the belief that a woman should tolerate some violence in her marriage to keep her family together [19]. Nationally representative surveys conducted in 2005 and 2016 also showed women accepting wife beating under certain conditions [5, 21]. Nevertheless, there was a positive change over time in the 2016 EDHS study, and our findings support that trend by providing contextual evidence that women in our study in 2022 stated that they were against any form of violence and normalization was not the case [5, 21].

Responses towards VAWG were reported as based on the perception or presumptions about violence. Several cases even if taken under the law were handled by customary mediation where the perpetrator did not face any punitive measures. Social pressure didn't prevent men from abusive acts but rather put all the blame on female violence survivors. Other studies in Ethiopia documented similar findings where violence acts were treated traditionally and women were advised to maintain their relationships by being submissive to their partners [19, 24]. Evidence also shows that customary mediation fails to recognize elements of human rights. The main disadvantage of such practices is that they overlook the damage and don't include women as part of the mediation which can lead to more stigmatization [9, 10]. As shown by several studies in Ethiopia, family responses towards the NPSV were still inclined to follow the cultural norms to forcing the survivor to marry the perpetrator [10, 25].

According to the world health report on violence and health, the different level factors in the SEM increase the risk of a women becoming a victim and a man becoming a perpetrator of violence [3]. This study also found that both IPV and NPSV are the result of multiple factors which range from individual to societal level. But most importantly the injustice in the legal system, the presence of customary mediation, the restrictive behavior parents showed towards their daughters, the lack of support from the community, presence of marked risky areas were perceived as contributor factor to rise VAWG in the community.

Our study found that criteria set for some forms of violence by the community, shaped violence survivors' and families' source of help. For instance, 'minor' and 'tolerated' forms of violence are reported to customary mediation, while 'unaccepted' violence is addressed legally. Some existing response services were reported as not responsive enough for the violence survivor. Survivors were challenged in the course of seeking help, leading them to seek help from families or friends, as shown in many studies in Ethiopia [5, 26, 27]. All informed families and friends were not resourceful enough to help them recover from their trauma or get proper care. Yet some directed them to appropriate formal services such as women's and children's

affairs office, lawyers, and the police. In line with this, the majority of violence survivors that we identified, from the narratives during group discussions, interviews and observations (beyond those we could interview in the study) were married women, domestic workers, internal migrants, sex workers, those who are less than 18 years old. All of them shared the same vulnerability factor of being economically disadvantaged. Even if the narratives revealed there were more cases of IPV, the care-seeking experience among them was less, this is consistent with other studies in Ethiopia [28].

According to the needs of violence survivors, disclosure and continuous emotional support from family, neighbors, community and legal sanctions were most useful to ensure continuity of psychological support. Late disclosure has consequences, such as not being able to access the right care at the right time. Studies showed that early disclosure is the first step to seeking help and a channel of escaping concurrent violence perpetration and other negative health outcomes [29]. Pertaining to prevention, awareness creation, building good parent-child communication, economic support, and strengthening religious organization's teaching were also reported to be important in preventing violence in the community. Evidence from a study in West Ethiopia showed that engaging religious leaders in violence prevention intervention brought positive change [23, 30]. A common belief and suggestion among men participants was that women have the power to stop every violence. This finding aligns with a qualitative study among university students in Ethiopia [28].

## Study strengths and limitations

We used triangulation of data from different methods to enhance the validity of our findings. Including violence survivors as participants and other individuals connected to VAWG services is a strength of the study. Our study limitations include the focus on one particular kebele, which limits the generalizability of the findings. Information bias is possible as few study participants had prior knowledge of VAWG. We did not conduct respondent validation of our interpretation of the findings, to validate the accuracy of our interpretation. Lastly, the number of violence survivors interviewed was limited based on the number of study participants at the care centers, and were unable to interview IPV survivors, which would have added more information on the care-seeking gap.

## Conclusion

Information pertaining to the study context about the perception, contributing factors, responses, and challenges to VAWG adds to the existing evidence on VAWG. The study concludes that VAWG programs should target audiences, and address different forms of violence through integration and collaboration between different level cadres. More importantly, primary prevention services focusing on gendered norms, gender equality, and awareness creation should be a priority in both urban and rural settings. Response interventions should include services that enhance and facilitate safe environments for women and girls to exercise their rights freely and ensure a continuum of care.

## Supporting information

**S1 Annex.**
(DOCX)

**S1 File. Description of key thematic areas.**
(DOC)

**S2 File. Inclusivity in global research.**
(DOCX)

**S3 File. List of existing services.**
(DOC)

## Acknowledgments

Our deepest gratitude goes to Southern Nations Nationalities and Peoples Region, Regional Health Bureau management, and the Ethical Review Committee for facilitating this study. Our special thanks also goes to Gamo zonal health department, Arbaminch town gender department, Arbaminch General Hospital, the one-stop center, and safe house managers and coordinators for their coordination and support. Our respect and appreciation goes to all respondents in this study. Lastly, we would like to thank Peter Macharia for drafting the map of the study site.

## Author Contributions

**Conceptualization:** Metasebia Admassu, Lenka Benova, Christiane Nöstlinger, Marie Laga, Thérèse Delvaux.

**Data curation:** Metasebia Admassu.

**Formal analysis:** Metasebia Admassu, Aline Semaan, Aliki Christou, Claudia Nieto-Sanchez.

**Funding acquisition:** Marie Laga.

**Investigation:** Metasebia Admassu, Misganu Endriyas.

**Methodology:** Metasebia Admassu.

**Supervision:** Metasebia Admassu, Thérèse Delvaux.

**Visualization:** Metasebia Admassu, Aline Semaan.

**Writing – original draft:** Metasebia Admassu.

**Writing – review & editing:** Metasebia Admassu, Lenka Benova, Christiane Nöstlinger, Aline Semaan, Aliki Christou, Claudia Nieto-Sanchez, Misganu Endriyas, Thérèse Delvaux.

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
