## [Decision Letter · Decision Letter 0]

10 Jan 2024

PONE-D-23-21320Uncovering community needs regarding gender-based violence among women and girls in southern Ethiopia: an explorative studyPLOS ONE

Dear Dr. Joffe,

Thank you for submitting your manuscript to PLOS ONE. After careful consideration, we feel that it has merit but does not fully meet PLOS ONE’s publication criteria as it currently stands. Therefore, we invite you to submit a revised version of the manuscript that addresses the points raised during the review process.

We look forward to receiving your revised manuscript.

Kind regards,

Vanessa Carels

Staff Editor

PLOS ONE

Journal Requirements:

"The authors declare that they have no competing interests."

5. We note that you have indicated that there are restrictions to data sharing for this study. PLOS only allows data to be available upon request if there are legal or ethical restrictions on sharing data publicly. For more information on unacceptable data access restrictions, please see http://journals.plos.org/plosone/s/data-availability#loc-unacceptable-data-access-restrictions. 

6. Please amend the manuscript submission data (via Edit Submission) to include authors Lenka Benova, Christiane Nöstlinger, Aline Semaan, Aliki Christou, Claudia Nieto-Sanchez, Marie Laga, Misganu Endriyas and Thérèse Delvaux.

7. We note that you have referenced (Southern Nation Nationalities People’s Region health bureau. Gender Based Violence Report /Unpublished/. 2021. ) which has currently not yet been accepted for publication. Please remove this from your References and amend this to state in the body of your manuscript: (ie “Bewick et al. [Unpublished]”) as detailed online in our guide for authors

8. We note that Figure 1 in your submission contain map/satellite images which may be copyrighted. All PLOS content is published under the Creative Commons Attribution License (CC BY 4.0), which means that the manuscript, images, and Supporting Information files will be freely available online, and any third party is permitted to access, download, copy, distribute, and use these materials in any way, even commercially, with proper attribution. For these reasons, we cannot publish previously copyrighted maps or satellite images created using proprietary data, such as Google software (Google Maps, Street View, and Earth). For more information, see our copyright guidelines: http://journals.plos.org/plosone/s/licenses-and-copyright.

Reviewers' comments:

Reviewer's Responses to Questions

**Comments to the Author**

1. Is the manuscript technically sound, and do the data support the conclusions?

Reviewer #1: Yes

Reviewer #2: Yes

2. Has the statistical analysis been performed appropriately and rigorously? 

Reviewer #1: N/A

Reviewer #2: Yes

3. Have the authors made all data underlying the findings in their manuscript fully available?

Reviewer #1: No

Reviewer #2: Yes

4. Is the manuscript presented in an intelligible fashion and written in standard English?

Reviewer #1: Yes

Reviewer #2: Yes

5. Review Comments to the Author

Reviewer #1: the manuscript is well prepared and clearly stated the findings of the research

regarding with data availability the author states there is some restriction for the data availability but the Plos one's data policy states the findings described fully available, without restriction, and from the time of publication

my questions

* I am not clear why you select women 15-49 years since violence occurs at all age category?

*how did you select kebele 6 for data collection is there any reason ? is it random selection or there are more GBV occurred in this kebele ?

*while you collect data from violence survivors you used informal conversation why? I think it is better to use in-depth interview

since your study title is focused on gender based violence why did you not include male participants who face violence? unless the title should be violence against women (VAW)

Reviewer #2: I appreciate the researchers' efforts in producing this thought-provoking paper. I had a great time reading this paper, and I've included a few remarks below.

1. Try to illustrate in the Introduction section how common gender-based violence (GBV) is around the world and how it differs in developing and non-developing nations. Make an effort to highlight any methodological or other shortcomings found in earlier research.

2. Under the methodology section, it is helpful to discuss the factors while selecting the research site (Arbaminch City, Gamo Zone, SNNPR).

6. PLOS authors have the option to publish the peer review history of their article (what does this mean?). If published, this will include your full peer review and any attached files.

Reviewer #1: No

Reviewer #2: **Yes: **Jibril Bashir Adem

---

## [Author Response · Author response to Decision Letter 0]

8 Mar 2024

Response to Academic Editors

1. Please ensure that your manuscript meets PLOS ONE's style requirements, including those for file naming

R: Thank you for your suggestions. We have now made all necessary changes according to the requirements of PLOS One.

2. Please include a complete copy of PLOS’ questionnaire on inclusivity in global research in your revised manuscript

R: Thank you for the suggestion. The “inclusivity in global research” is now added to the methods section. Please see on page “7”.

3. We note that the grant information you provided in the ‘Funding Information’ and ‘Financial Disclosure’ sections do not match

R: Thank you for the comment. We have now aligned information in the “Funding Information” and “Financial Disclosure” sections. 

4. Please complete your Competing Interests on the online submission form to state any Competing Interests

R: Thank you for the comment. We have completed the competing interest section on the online submission form. 

5. We note that you have indicated that there are restrictions to data sharing for this study. PLOS only allows data to be available upon request if there are legal or ethical restrictions on sharing data publicly

R: Thank you. Yes, we agree with PLOS one recommendations. Due to the sensitive nature of the topic, ethical requirements, and data protection rules at the Institute of Tropical Medicine, Antwerp, we cannot make data publicly available. However, a reasonable request can be sent to the corresponding author who will verify the request and pass it to the data repository unit of the Institute for final approval and data sharing. 

6. Please amend the manuscript submission data (via Edit Submission) to include authors Lenka Benova, Christiane Nöstlinger, Aline Semaan, Aliki Christou, Claudia Nieto-Sanchez, Marie Laga, Misganu Endriyas and Thérèse Delvaux

R: Thank you. We have amended the manuscript submission data accordingly. 

7. We note that you have referenced (Southern Nation Nationalities People’s Region health bureau. Gender-Based Violence

R: Thank you for the suggestion. We have amended the required changes to the manuscript. 

8. We note that Figure 1 in your submission contains map/satellite images which may be copyrighted. All PLOS content is published under the Creative Commons Attribution License (CC BY 4.0), which means that the manuscript, images, and Supporting Information files will be freely available online, and any third party is permitted to access, download, copy,

distribute, and use these materials in any way, even commercially, with proper attribution. For these reasons, we cannot publish previously copyrighted maps or satellite images created using proprietary data, such as Google software (Google Maps, Street View, and Earth)

R: Thank you for the remark.

The authors created Figure 1, which is not based on satellite data. The author created the figure using ArcMap V.10.5 (ESRI, Redlands, California, USA) with data from OpenStreetMap. OpenStreetMap® is open data, licensed under the Open Data Commons Open Database License (ODbL) by the OpenStreetMap Foundation (OSMF). In the license, users are free to copy, distribute, transmit, and adapt OSM data, as long as OpenStreetMap and its contributors are credited, which we have done. 

We refer the editor to the detailed version of this from OSM at this link: https://www.openstreetmap.org/copyright#:~:text=OpenStreetMap%C2%AE%20is%20open%20data,credit%20OpenStreetMap%20and%20its%20contributors. 

9. Please review your reference list to ensure that it is complete and correct. If you have cited papers that have been retracted, please include the rationale for doing so in the manuscript text, or remove these references and replace them with relevant current references. Any changes to the reference list should be mentioned in the rebuttal letter that accompanies your revised manuscript. If you need to cite a retracted article, indicate the article’s retracted status in the

References list and also include a citation and full reference for the retraction notice

R: Thank you. We have reviewed the references checked for incompleteness and made necessary changes. We have not cited retracted papers.

Response to Reviewer 1

1. I am not clear why you select women 15-49 years since violence occurs at all age category?

R: Thank you for the comment.

We agree, violence can happen at any age. However, we are focusing on the two forms of violence, which are intimate partner violence (IPV) and non-partner sexual violence (NPSV). By definition of IPV and NPV, for both forms of violence, the inclusion criteria starts at age 15, because this is the minimum age to give consent to participate in a study in Ethiopia. The age group considered to be at the highest risk for IPV and NPSV is between 15-49 years, which is also the reproductive age group. We included young women aged 15-18 in our study only if they were emancipated minors (i.e. working to earn their living, married, parenting) as per the Ethics guideline in Ethiopia, and could legally consent to participate in the study.

The definitions can be obtained via this link: 

9789241564625_eng.pdf (who.int) 

Link to the Ethiopa National Research Ethics Review Guideline can be obtained via this link:

https://www.merlot.org/merlot/viewMaterial.htm?id=772743972

2. How did you select kebele 6 for data collection is there any reason ? is it random selection or there are more GBV occurred in this kebele ?

R: Thank you for the question. 

We selected kebele 6 purposely as the site has a higher number of reported cases of VAWG than the rest of the kebeles in the zone during the time of data collection. We have included the justification for selecting kebele 6 in our methods part (participant and site selection, Page 5).

3. While you collect data from violence survivors you used informal conversation why? I think it is better to use in-depth interview

R: Thank you for the suggestion.

We used three different methods of data collection (in-depth interviews, informal conversations, and observations). We conducted in-depth interviews with three violence survivors at the safe house and 1 informal conversation at the one-stop center (please refer to Table 1). So for IDIs with violence survivors, our site of recruitment was only the safe house. However, during the time of data collection, the number of violence survivors was minimal, and those who were willing to participate in the study were only three. As per our protocol, we did IC with violence survivors at the one-stop center to increase the internal validity of the data collected. Please refer to method part page 5, under the “participant selection section”.

4. Since your study title is focused on gender-based violence why did you not include male participants who face violence? unless the title should be Violence Against Women (VAW)

R: Thank you for the suggestion. We preferred to use gender-based violence as most studies use them interchangeably. However, we agree with the suggestion, that it should be changed to VAWG. Please see the change in the title and all the contents in the document. 

Response to Reviewer 2

1. Try to illustrate in the Introduction section how common gender-based violence (GBV) is around the world and how it differs in developing and non-developing nations. Make an effort to highlight any methodological or other shortcomings found in earlier research

R: Thank you for your review and feedback. 

We have added the suggested points regarding how common GBV is around the world and how it differs in developing and non-developing nations to the introduction part. Please see page 3. 

More importantly, we were convinced to conduct this study as context-specific research is valuable to develop locally adapted solutions to local problems, which we have described in the last part of the introduction.

2. Under the methodology section, it is helpful to discuss the factors while selecting the research site (Arbaminch City, Gamo Zone, SNNPR).

R: Thank you for the feedback. 

We selected Arbaminch purposively as the prevalence of gender-based violence was higher around the area according to the report of the regional health bureau (Southern Nations Nationalities and Peoples’ Regional Health Bureau annual report, 2021 [unpublished]). We also selected “Kebele 6” purposely based on the higher number of GBV survivors registered at the one-stop center. The justification is now included in the methods section under the participant and site selection on page 5.

---

## [Editor Report · Decision Letter 1]

14 May 2024

Uncovering community needs regarding violence against women and girls in Southern Ethiopia: an explorative study

PONE-D-23-21320R1

Dear Dr. Joffe,

We’re pleased to inform you that your manuscript has been judged scientifically suitable for publication and will be formally accepted for publication once it meets all outstanding technical requirements.

Kind regards,

Ranjan Kumar Prusty, Ph.D.

Academic Editor

PLOS ONE
---

## [Editor Report · Acceptance letter]

31 May 2024

PONE-D-23-21320R1 

PLOS ONE

Dear Dr. Joffe, 

I'm pleased to inform you that your manuscript has been deemed suitable for publication in PLOS ONE. Congratulations! Your manuscript is now being handed over to our production team.

Kind regards, 

on behalf of

Dr. Ranjan Kumar Prusty 

Academic Editor

PLOS ONE